# Towards the Essence of Progressiveness: Bringing Progressive Fibrosing Interstitial Lung Disease (PF-ILD) to the Next Stage

**DOI:** 10.3390/jcm9061722

**Published:** 2020-06-03

**Authors:** Laurens J. De Sadeleer, Tinne Goos, Jonas Yserbyt, Wim A. Wuyts

**Affiliations:** 1Laboratory of Respiratory Diseases and Thoracic Surgery (BREATHE), Department CHROMETA, KU Leuven, B-3000 Leuven, Belgium; laurens.desadeleer@kuleuven.be (L.J.D.S.); tinne.goos@kuleuven.be (T.G.); jonas.yserbyt@uzleuven.be (J.Y.); 2Unit of Interstitial Lung Diseases, Department of Respiratory Diseases, University Hospitals Leuven, B-3000 Leuven, Belgium

**Keywords:** progressive fibrosing interstitial lung disease, pulmonary fibrosis, interstitial lung disease

## Abstract

Although only recently introduced in the ILD community, the concept of progressive fibrosing interstitial lung disease (PF-ILD) has rapidly acquired an important place in the management of non-idiopathic pulmonary fibrosis fibrosing ILD (nonIPF fILD) patients. It confirms a clinical gut feeling that an important subgroup of nonIPF fILD portends a dismal prognosis despite therapeutically addressing the alleged triggering event. Due to several recently published landmark papers showing a treatment benefit with currently available antifibrotic drugs in PF-ILD patients, endorsing a PF-ILD phenotype has vital therapeutic consequences. Importantly, defining progressiveness is based on former progression, which has proven to be a rather moderate predictor of future progression. As fibrosis extent >20% and the presence of honeycombing have superior predictive properties regarding future progression, we advocate immediate initiation of antifibrotic treatment in the presence of these risk factors. In this perspective, we describe the historical context wherein PF-ILD has emerged, determine the currently employed PF-ILD criteria and their inherent limitations and propose new directions to mature its definition. Finally, while ascertaining progression in a nonIPF fILD patient clearly demonstrates the need for (additional) therapy, in the future, therapeutic decisions should be taken after assessing which pathway is ultimately driving the progression. Although not readily available, pathophysiological insight and diagnostic means are emergent to go full steam ahead in this novel direction.

## 1. Introduction

The conceptualization of the existence of a common progressive fibrosing (PF) phenotype, irrespective of the underlying diagnostic entity, has dramatically changed the landscape of interstitial lung disease (ILD) [1]. In this concept, patients with a fibrosing ILD who have shown to experience disease progression evidenced by pulmonary function decline are lumped into one group, irrespective of the underlying diagnostic entity. The impact of this concept and its associated transformation of PF-ILD treatment will probably approximate the same order of magnitude as the paradigm shift induced by the PANTHER-IPF trial [2] in idiopathic pulmonary fibrosis (IPF). As this trial showed manifest deleterious effects of immunosuppression in IPF, the standard of care dramatically changed, and immunosuppression was totally abandoned. This perspective evaluates the current definition of this phenotype and discusses proper evolution towards the next stage in assessing progression in fibrosing ILD patients.

## 2. Historical Context

The origins of the paradigm shift in pulmonary fibrosis are situated in the early years of this millennium, in which it became clear that IPF was not initiated by an exaggerated immune response that could serve as a treatable target, but should be regarded as a process of self-sustaining fibrosis [3], without an identifiable cause or trigger. This model rapidly became the generally accepted narrative of IPF pathophysiology after the publication of the PANTHER-IPF trial [2], revealing that immunosuppressive treatment resulted in increased mortality without attenuating pulmonary function decline. As both Nintedanib and Pirfenidone proved to be an effective means of reducing IPF progression [4,5,6], it became clear that fibrosis was manageable by targeting the fibrotic mechanisms itself, rather than focusing on possible (early) triggering events.

Based on these seminal findings, a dichotomy in fibrotic ILD found entrance: IPF was regarded as an intrinsically fibrotic disease, whereas, in nonIPF fibrotic ILDs (fILD), one should focus on the initial trigger (i.e., an underlying connective tissue disease (CTD) in CTD-associated ILD (CTD-ILD), an inciting agent in fibrotic hypersensitivity pneumonitis (fHP) and a presumed immunological reaction in idiopathic nonspecific interstitial pneumonia (NSIP)) and treat accordingly, which—in daily clinical practice—was translated almost invariably to the initiation of immunosuppressive therapies. Given the lower mortality levels and known triggers, these nonIPF fILD entities were regarded as more benign, largely treatable and oftentimes reversible.

Throughout the following years, a growing body of evidence strengthened an emerging clinical gut feeling that some subgroups within the spectrum of nonIPF fILD showed a progressive fibrosis that did not seem to resolve or stabilize by therapeutically addressing the alleged initial trigger [7,8,9,10,11]. More cohort studies were published showing the rather disappointing treatment results of immunosuppression in different groups of nonIPF ILD [12,13,14]. As some of these patients had remarkable similarities with regard to computed tomography (CT) pattern or histopathological findings, lively discussion arose in many multidisciplinary meetings about whether these patients should be diagnosed with a highly fibrotic nonIPF fILD (in which no other treatment than immunosuppression was available) or with a more atypical IPF (which would result in the possibility of starting antifibrotic treatment). As more studies reported shared pathophysiological mechanisms between IPF and nonIPF fILD, predisposing to inferior outcomes [1,15,16,17], the idea grew that antifibrotic drugs might equally mitigate disease progression in nonIPF fILD. Moreover, the concept of disease behavior was well established by that time [18] and was a welcomed confirmation within the ILD community that immunosuppression-irresponsive progressively fibrosing nonIPF patients did exist and uncovered the high medical need for new treatment options. Although hints and broad concepts were suggested, the classification of such disease behavior was rather dependent on clinical acumen. At present, robust data are still lacking regarding the proportion of nonIPF fILD patients who eventually experience significant and clinically relevant progressive fibrosis. A survey study involving 243 pulmonologists, 203 rheumatologists and 40 internists showed that the community estimates this proportion at 18–32%, although clinical cohort studies hint that it is around 50% [9,10].

Unfortunately, the relatively low prevalence of separate nonIPF fILD entities complicated the options for testing the efficacy of the antifibrotics in clinical trials. Given the plethora of therapeutic targets of both antifibrotics, the idea grew that while the exact contribution of specific molecular mechanisms in the progression of fibrosis might differ from one nonIPF fILD to another, these drugs could attenuate the progression of all nonIPF fILD patients [19]. This hypothesis has been tested in multiple trials, reporting beneficial outcomes in 2019. Both Pirfenidone and Nintedanib were shown to reduce forced vital capacity (FVC) decline in PF-ILD cohorts [20,21] as well as progressive unclassifiable ILD (UILD) patients [22].

Importantly, the benefits of clinically defining progressive fibrosis reach far wider than evaluating potential antifibrotic responsiveness. We envisage a role in referral for lung transplantation and/or the initiation of advanced care planning and initiation of palliative care. Moreover, as 50% of IPF patients die from cardiovascular comorbidities, cardiovascular prevention measures could be useful if the comorbidome of PF-ILD would prove to be similar to that of IPF. Furthermore, from a research perspective, defining patients with progressive fibrosis could be very useful as a patient group of specific interest.

## 3. Defining Progressiveness: From Clinical Trial Endpoint to Clinical Practice

### 3.1. Progressive Fibrosis: Clinical Trial Endpoint

In order to selectively include nonIPF fILD patients with progressive fibrosis, the clinical trials assessing the efficacy of antifibrotic treatment in nonIPF PF-ILD selected patients with proven former disease progression throughout the months before study inclusion.

Progression was defined slightly differently in each study. The RELIEF study [20], initiated by the German Lung Research Consortium, assessed the efficacy of Pirfenidone in PF-ILD. Progression was defined as an annualized FVC decline of at least 5% per year, within an interval of 0.5–2 years. Another study, assessing the efficacy of Pirfenidone in progressive fibrosing unclassifiable ILD [22], defined progression as a 5% absolute decline in FVC (% predicted) or a significant symptomatic worsening not due to cardiac, pulmonary (except worsening of underlying unclassifiable ILD), vascular or other causes within 6 months. In the INBUILD trial [21], which evaluated the efficacy of Nintedanib in PF-ILD, progression was defined as one of the following within 24 months: FVC% decline >10% or two elements of the following: FVC% decline between 5% and 10%, symptom worsening or an increased extent of high-resolution CT (HRCT)-derived fibrosis. Recently, Brown et al. reported that patients included in the placebo group of the INBUILD had similar clinical behavior during the trial in terms of FVC decline and mortality compared to the placebo groups of the INPULSIS trial, which evaluated Nintedanib in IPF [23], confirming that PF-ILD is a dismal phenotype to be regarded as equally grave compared to IPF.

Although these criteria certainly are not without merit, we believe that defining progression based on these criteria might have certain limitations.

Firstly, by definition, the patient and clinician should wait until deterioration has been formally determined before the progressive fibrotic phenotype can be endorsed. Hence, precious time is lost, and the disease has objectively worsened in the meantime. Until we have treatment options that can significantly reverse fibrosis, this is a significant problem. In IPF, it takes approximately 2 years from symptom onset to diagnosis, and a survey study recently reported similar estimated delays in nonIPF fILD [24]. One might believe that—given the presumed favorable disease behavior of nonIPF fILDs compared to IPF—this lag time will mostly compromise part of the pulmonary function reserve rather than mitigating essential functional pulmonary tissue and thus can be acceptable. Such a rationale would ignore two important issues. First, a recently published post-hoc analysis elegantly showed similar outcomes in the placebo groups of the INPULSIS and INBUILD trials, thus providing evidence that PF-ILD patients confer similar dismal outcomes compared to IPF patients. Secondly, one should keep in mind that—as nonIPF fILD patients are diagnosed at a far younger age compared to IPF—a similar post-diagnosis survival would imply a far higher loss in life-years in nonIPF patients compared to IPF patients. Loss of pulmonary function should be minimized, and thus, predicting the risk of progression upfront at diagnosis will prove to be essential.

Secondly, we believe the criteria might be far too strict for encompassing all progressive fibrosing nonIPF ILD patients in an acceptable time interval. Although IPF is invariably regarded as a relentlessly progressive disease, the placebo groups of the phase III trials evaluating the efficacy of the antifibrotics showed that only a minority of cases over the study period had such magnitude of a decline in pulmonary function, symptomatology or HRCT-derived fibrosis severity used for diagnosing the PF phenotype in the nonIPF PF-ILD trials. In the CAPACITY trial evaluating Pirfenidone versus placebo [4], only 32% of cases in the placebo group had an FVC decline >10% over the 72-week study period, and less than half had a 6-min walk decrement >50 m (which was defined as the cut-off for clinical relevance). In the ASCEND trial [5], these data were confirmed; moreover, the composite secondary endpoint of a significant increase in dyspnea scoring or death was reported in only 36% of placebo cases. In the INPULSIS trial assessing efficacy and safety of Nintedanib vs. placebo [6], these data were roughly replicated, and an FVC decline <5% was observed in the placebo groups in almost 40%. Moreover, the mean St. George’s Respiratory Questionnaire (SGRQ) symptoms domain decrement during the trial in the placebo group was 3.67 ± 0.94, which is lower than the minimal clinically important difference (MCID), indicating that at least an important minority of patients probably did not experience a meaningful increase in symptoms. This would mean that—if the IPF patients included in these trials had been screened for progressive fibrosis in an approach similar to that used in the INBUILD trial—roughly one-third would require a >1-year interval before progression could be objectivated and treatment could be initiated.

Thirdly, FVC might not be a good predictor in all fibrotic ILD patients. Many nonIPF fILD cases may present with a combination of emphysema and fibrosis (CPFE). In systemic sclerosis-associated ILD (Ssc-ILD), 7.8% of cases have been shown to present with CPFE [25], while almost 30% of rheumatoid arthritis-associated ILD (RA-ILD) cases have concomitant emphysema [26]. Only one-third of CPFE patients are estimated to have IPF [27]. In IPF patients presenting with a CPFE pattern, it has been shown that FVC decline is not a good marker of disease progression and only weakly associates with outcome [28,29]. Waiting for FVC decline in nonIPF fILD patients presenting with a CPFE pattern will result in woeful time loss in this subgroup of patients with a known dismal prognosis [30].

Finally, and most importantly, former progression proved to be a rather moderate predictor of future progression. Although all patients included in the INBUILD trial were progressive at inclusion (as this was needed for inclusion), only 40% and 60% of the placebo cases had a >10% and 5–10% FVC decline, respectively, throughout the first 52 weeks of the trial [23]. Hence, former progression predisposes a risk of only ±50% for future progression.

### 3.2. Progressive Fibrosis: Clinical Practice

Based on the limitations mentioned above, unraveling clinical parameters that can predict long-term disease progression will prove to be crucial. In the last 10 years, multiple cohort studies have been published in various nonIPF ILD entities, which can aid in this respect. In short, most studies reveal both baseline pulmonary function, HRCT parameters and multilevel composite scoring systems to be predictors of disease progression.

FVC has been associated with survival in UILD [9,10], fHP [8,31] and CTD-ILD [11,32]. Lower diffusing capacity of the lung for carbon monoxide (DLCO) conferred worse survival in UILD [9,10], fHP [8,31], CTD-ILD [11,32,33] and idiopathic NSIP [7]. Moreover, FVC and DLCO were associated with disease progression in UILD [9].

Key messages:The PF-ILD has been conceptualized as an important subgroup of nonIPF fILD patients portend dismal outcomes similar to IPF, despite therapeutically addressing the alleged triggering event.Clinical studies have shown antifibrotics to be effective in nonIPF fILD patients with former disease progression. However, former progression has proven to be a rather moderate predictor of future progression; moreover, loss of time and pulmonary function is inherent before the PF-ILD phenotype can be endorsed.Clinical variables (e.g., radiological disease extent and honeycombing presence) might perform better as predictor of progression and can be assessed at time of diagnosis.In the future, predicting progression will ultimately inform the need for (additional) treatment, while assessing the progression-driving disease mechanisms will guide treatment choices.

The extent of fibrosis, traction bronchiectasis and honeycombing have been shown to be associated with survival in UILD [9], fHP [8,31], CTD-ILD [33], RA-ILD [32] and Ssc-ILD [11]. Honeycombing presence conferred worse survival after correction for sex, age, FVC, DLCO, ILD subtype and immunosuppressive therapy in a broad group of fibrosing ILD patients [34]. The association between honeycombing and fibrosis extent with mortality was confirmed in two CPFE cohorts [27,35]. The extent of fibrosis was associated with disease progression in UILD [9], and the presence of honeycombing conferred a mean FVC decline of >5% [8].

In 2003, Wells and colleagues constructed the composite physiological index (CPI), an estimator of fibrosis extent on CT based on FVC, DLCO and forced expiratory volume in 1 s (FEV1), which proved to be associated with mortality [36]. CPI proved to be associated with survival in UILD [9], fHP [31], CTD-ILD [33] and RA-ILD [32]. Moreover, CPI was associated with disease progression in a UILD cohort [9]. Ley and colleagues developed a mortality prediction score in IPF using age, gender, FVC and DLCO, which was named the gender-age-physiology (GAP) system [37,38]. The system was expanded for other ILD entities by slightly modifying the scoring system into the ILD-GAP scoring [39]. ILD-GAP was associated with survival in CTD-ILD [33] and UILD [10], although GAP also encompasses variables such as age, which are linked with survival but not necessarily with disease progression. Interestingly, when IPF was part of the differential diagnosis (which is cumulative of a wide range of clinical clues), the odds ratio for disease progression was 5.5 in the Ryerson paper [10]. Interestingly, multivariate modeling showed that both DLCO% and fibrosis extent were independently associated with disease progression, which indicates that pulmonary function and CT variables might be complementary in predicting disease progression in nonIPF fILD.

Even more important than these reported hazards and odds ratios, absolute survival and disease progression rates are often reported and almost invariably show similarities with IPF.

Remarkably, based on the data provided in the Ryerson paper addressing outcome predictors in UILD [9], one can easily calculate that roughly 75% of UILD cases presenting with honeycombing will experience disease progression in the first year after evaluation, which is far more than the 1-year risk of disease progression in IPF, based on the data of the placebo groups in the phase III antifibrotics trials, outlined above. In other words, these data suggest that a UILD case with honeycombing would have a higher risk of disease progression than a random IPF case in the first year after diagnosis. Even more importantly, as mentioned before, from all patients included in the placebo arms of the INBUILD trial (thus with evidence of progression in the previous months), only 40–60% experienced further progression throughout the 52 weeks of the trial. Hence, former disease progression conferred a lower risk for future progression compared to the presence of honeycombing.

This holds true for many other nonIPF fILD subgroups. Salisbury showed that HP with honeycombing conferred a similar outcome (with regard to both survival and FVC decline) to that of IPF with honeycombing and had a worse survival and FVC decline compared to IPF cases without honeycombing. Jacob showed that RA-ILD cases with honeycombing, irrespective of the distribution, showed similar outcomes (i.e., 48% 3-year survival rate) compared to an IPF cohort (42% 3-year survival rate). It is by no means justifiable that in such a patient subgroup, accounting for 25–50% of nonIPF fILD cases [8,9,11,32,34], disease progression should be awaited before initiation of antifibrotics.

The very same case can be made for fibrosis extent. Ryerson showed a 3-year survival of nearly 50% in UILD cases with fibrosis extent >20% of the lung parenchyma (i.e., reticulation plus honeycombing) [9]. Jacob et al. showed that survival in RA-ILD with fibrosis extent >25% (or fibrosis extent in between 15% and 25% in conjunction with an FVC% <70%) was similar compared to IPF (i.e., median survival of around 3 years) [32].

In conclusion, whereas both pulmonary function variables and CT variables and multilevel composite scoring systems are all associated with mortality, the presence of honeycombing and a CT-derived fibrosis extent >20% are clearly associated with disease progression and IPF-compatible outcomes. Hence, we believe that enough evidence exists to justify the immediate initiation of antifibrotic treatment.

## 4. What Makes the Progressive Phenotype Progressive? The Quest for the Underlying Molecular Mechanisms

Whereas predicting clinical behavior will clarify whether additional therapeutic action is needed, unraveling the underlying molecular mechanisms that drive disease progression will inform which therapy might be useful.

Given the broad range of mechanisms that are therapeutically targeted with the currently available antifibrotic drugs and given the absence of other drugs that work crucially differently, the question of which drug should be initiated in a specific patient is—at present—without much meaning. However, as insight into the pathogenesis of both IPF and nonIPF fILD is rapidly accrued, rather soon than late, clinicians will need to make a well-informed choice about which mechanism he or she believes is driving the disease progression preponderantly, and thus which treatment option—yet to be discovered—to initiate primarily.

While fibrogenesis is often abusively confined to extra-cellular matrix deposition, abundances of other mechanisms have been revealed to be pathogenetically substantial and are presumably not responsive to the currently available antifibrotic drugs. In IPF, epithelial senescence is a widely accepted early phenomenon [40,41], including mitochondrial dysfunction [42], leading to a failing regenerative response to repetitive micro-injuries, oftentimes induced by environmental exposures. Moreover, the emergence of an aberrant airway basal cell population has been widely acknowledged [43,44], as well as reduced angiogenesis [45]. The early involvement of small airways has recently been discovered [46], and the important role of airway malformation and development of honeycombing [47] has been widely adopted, as well as the enigmatic influence of MUC5B polymorphisms and the resulting dysfunctional mucociliary clearance [48,49,50]. Recently, it has been suggested that microbiome changes might prove to be an important factor [51,52], irrespective of baseline disease severity [53]. Finally, the profibrotic role of macrophages with an M2 phenotype is established [54,55].

While these pathways do not reach the full attention of the ILD clinician, as no targeting treatment is available, trials are underway to investigate the potential impact on the progression of fibrosis: thyroid hormone has been suggested for mitigating mitochondrial dysfunction [56], Dasatinib-Quercitin, which inhibits the senescence-associated secretory phenotype of senescent cells, is currently being tested in IPF [57,58], potential therapies mitigating MUC5B-driven impaired mucociliary clearance are suggested [50], and even type II alveolar epithelial cell transplantations are being tested [59]. Moreover, a trial testing antimicrobial agents is being set up [60], and targeting gastro-esophageal reflux disease has recently shown some beneficial results [61]. Finally, mitigating the differentiation from monocytes into M2 macrophages by recombinant human pentraxin-2 shows a very promising therapeutic option [62].

As the clinical behavior of PF-ILD patients has proven to have so many aspects of IPF in common, it would be very surprising if these novel IPF-driving mechanisms did not have a role in nonIPF fILD disease progression. The first results in this regard have been published within the past few years [15,16,41,63]. More than guiding treatment based on underlying diagnostic entity, the future’s choices in fILD management probably will be made based on underlying prevailing activated mechanisms, irrespective of the ILD entity.

We believe that—in the future—disease progression will be ascertained by assessing aberrant activation of pathophysiological mechanisms. Biomarkers will be needed for this purpose, and the technology for development is about to be mature [43,64,65,66] in terms of (epi-)genetic testing (e.g., MUC5B genotype and telomere length) and molecular classifiers.

## 5. Conclusions

Throughout the last decennium, the paradigm shift in the IPF pathophysiological narrative and the development of effective antifibrotic drugs has proven to be an important impetus for studying disease progression in nonIPF fILD, resulting in seminal phase III trials showing that antifibrotics are as effective in PF-ILD as they are in IPF. Former progression was used as inclusion criteria for these trials, but important limitations hamper their clinical use in practice: their ability to predict future progression was rather moderate, and the fact that disease progression should be awaited before treatment can be initiated might be unjustifiable in some patient subgroups: the presence of disease extent >20% or honeycombing was shown to predict future progression at least as well as former progression. Hence, treatment should be initiated as soon as possible, and disease progressions should not be awaited. Finally, as a more diverse collection of pathophysiological mechanisms are demonstrated to drive fibrosis, the future might bear more personalized medicine, in which disease progression will be predicted by biomarkers that ascertain aberrant activation of specific pathways and steer therapeutic choices alike.

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
