# Peer review of "Towards the Essence of Progressiveness: Bringing Progressive Fibrosing Interstitial Lung Disease (PF-ILD) to the Next Stage"

_jcm, 2020, doi:10.3390/jcm9061722_

Round 1

Reviewer 1 Report

The authors have provided an interesting perspective on the history and the reasons behind the shift the fILD field has gone through recently. The authors show that the new concept that has emerged will have a significant impact on the field which I agree upon.

I feel however some minor improvements could be made.

The abstract and introduction are very much written for experts in the lung(fibrosis) field and given the broad audience of the Journal of Clinical Medicine too difficult to read. The abstract and introduction should therefore be revised to make it suitable for a broader audience. For example, what is the ‘concept of progressive fibrosing interstitial lung disease’ the authors mention in the first sentence. Also many readers might not be familiar with the PANTHER-trial and will miss the message of the paper.

The most important conclusion made by the authors is that the “presence of HC and >20% CT-derived fibrosis justifies immediate initiation of antifibrotic treatment” (line 213/214). This conclusion should therefore be included in the abstract. Additionally, the current conclusion that personalized medicine is almost available (“are about to be at our disposal”) is maybe a bit too far-fetched and makes the perspective also a little irrelevant. Why would we still bother to treat all non-IPF fILD and IPF the same way?

In the first part of chapter 3 the authors have not included the SENSCIS trial of which the results were released at the same time as the INBUILD trial looking at progressive lung fibrosis in systemic sclerosis patients, whereas they do mention this entity further down in this chapter. The authors should include this trial in the analysis like they have done for the pirfinedone and nintedanib trials in progressive lung fibrosis.

The amount of abbreviations makes it difficult to read for laymen and a table/list of all abbreviations used would be helpful. Also the abbreviation for honeycombing is not explained properly (used for the first time in line 166) and not consistently either (in line 169 the full word is used again).

There are some minor grammatical/spelling errors throughout the manuscript. For instance:
- line 169: ‘conferred a mean’ instead of ‘conferred an mean’.
- line 199: ‘as’ instead of ‘than’.
- line 245: ‘shown’ instead of ‘showed’.

Author Response

The authors have provided an interesting perspective on the history and the reasons behind the shift the fILD field has gone through recently. The authors show that the new concept that has emerged will have a significant impact on the field which I agree upon.

I feel however some minor improvements could be made.

The abstract and introduction are very much written for experts in the lung(fibrosis) field and given the broad audience of the Journal of Clinical Medicine too difficult to read. The abstract and introduction should therefore be revised to make it suitable for a broader audience. For example, what is the ‘concept of progressive fibrosing interstitial lung disease’ the authors mention in the first sentence. Also many readers might not be familiar with the PANTHER-trial and will miss the message of the paper.

We thank the reviewer for his interesting comment. We modified the abstract and drastically enlarged the introduction, providing some basic information regarding the concept of PF-ILD.

Abstract:

“Although only recently introduced in the ILD community, the concept of progressive fibrosing interstitial lung disease (PF-ILD) has rapidly acquired an important place in the management of non-idiopathic pulmonary fibrosis fibrosing ILD (nonIPF fILD) patients. It confirms clinical gut feeling that an important subgroup of nonIPF fILD portend a dismal prognosis despite therapeutically addressing the alleged triggering event. Due to several recently published landmark papers showing treatment benefit with currently available anti-fibrotic drugs in PF-ILD patients, endorsing a PF-ILD phenotype has vital therapeutic consequences. Importantly, defining progressiveness is based on former progression which has proven to be a rather moderate predictor of future progression. As fibrosis extent >20% and the presence of honeycombing have superior predictive properties regarding future progression, we advocate immediate initiation of antifibrotic treatment in the presence of these risk factors. In this perspective, we describe the historical context wherein PF-ILD has emerged, determine the currently employed PF-ILD criteria & their inherent limitations and propose new directions to mature its definition. Finally, while ascertaining progression in a nonIPF fILD patient clearly demonstrate the need for (additional) therapy, in the future therapeutic decisions should be taken after assessing which pathway is ultimately driving the progression. Although not readily available, pathophysiological insight and diagnostic means are emergent to go full steam ahead in this novel direction.”

Introduction:

“The conceptualization of the existence of a common progressive fibrosing (PF) phenotype, irrespective of the underlying diagnostic entity has dramatically changed the landscape of interstitial lung disease (ILD) [1]. In this concept, patients with a fibrosing ILD who have shown to experience disease progression evident by pulmonary function decline, are lumped into one group, irrespective of the underlying diagnostic entity. Probably, the impact of this concept and its associated transformation of PF-ILD treatment will approximate the same order of magnitude as the paradigm shift induced by the PANTHER-IPF trial [2] in idiopathic pulmonary fibrosis (IPF). As this trial showed manifest deleterious effects of immunosuppression in IPF, standard of care dramatically changed, and immunosuppression was totally abandoned. This perspective will evaluate the current definition of this phenotype and discuss proper evolutions towards a next stage in assessing progression in fibrosing ILD patients. “

The most important conclusion made by the authors is that the “presence of HC and >20% CT-derived fibrosis justifies immediate initiation of antifibrotic treatment” (line 213/214). This conclusion should therefore be included in the abstract.

We acknowledge that this message should be included in the abstract. We added:

“Moreover, as fibrosis extent >20% and the presence of honeycombing have superior predictive properties regarding future progression, we advocate immediate initiation of antifibrotic treatment in the presence of these risk factors.”

Moreover, we added a Panel with key messages:

  • “The PF-ILD has been conceptualized as an important subgroup of nonIPF fILD patients portend dismal outcomes similar to IPF, despite therapeutically addressing the alleged triggering event.
  • Clinical studies have shown antifibrotics to be effective in nonIPF fILD patients with former disease progression. However, former progression has proven to be a rather moderate predictor of future progression; moreover, loss of time and pulmonary function is inherent before the PF-ILD phenotype can be endorsed.
  • Clinical variables (e.g. radiological disease extent and honeycombing presence) might perform better as predictor of progression and can be assessed at time of diagnosis.
  • In the future, predicting progression will ultimately inform the need for (additional) treatment, while assessing the progression-driving disease mechanisms will guide treatment choices.”

Additionally, the current conclusion that personalized medicine is almost available (“are about to be at our disposal”) is maybe a bit too far-fetched and makes the perspective also a little irrelevant. Why would we still bother to treat all non-IPF fILD and IPF the same way?

The central theme of the last part of the perspective is that whereas establishing disease progression will merely clarify whether additional therapy is needed, unravelling the underlying molecular mechanisms which drive disease progression will inform which therapy might be useful. Hence, treatment decisions will be taken based on the underlying prevailing pathways rather than based on the diagnostic ILD entity that was established. In other words, we need to lump all PF-ILD patients, in order to split them based on underlying drivers rather than based on ILD subtype.

We added to the last paragraph:

More than guiding treatment based on underlying diagnostic entity, the future’s choices in fILD management probably will be made based on underlying prevailing activated mechanisms, irrespective the ILD entity.”

In the first part of chapter 3 the authors have not included the SENSCIS trial of which the results were released at the same time as the INBUILD trial looking at progressive lung fibrosis in systemic sclerosis patients, whereas they do mention this entity further down in this chapter. The authors should include this trial in the analysis like they have done for the pirfinedone and nintedanib trials in progressive lung fibrosis.

While INBUILD selectively included patients with a former disease progression, inclusion for SENCIS did not require former disease progression. Only a disease extent on CT > 10% was required. Hence, we did not add the SENSCIS trial in this manuscript, as it is focused on PF-ILD, rather than on whether antifibrotics work or not. To the best of our knowledge, no post-hoc analyses of the SENSCIS trial were published to evaluate specific effects of Nintedanib in patients with former disease progression. As we did not describe the effects of Nintedanib in the INBUILD (and other PF-ILD trials) in detail, we think adding the results of the SENSCIS might not be relevant to the reader. We added the Goh study as this study evaluated risk factors for mortality in systemic sclerosis, which we used as a proxy for disease progression.

The amount of abbreviations makes it difficult to read for laymen and a table/list of all abbreviations used would be helpful. Also the abbreviation for honeycombing is not explained properly (used for the first time in line 166) and not consistently either (in line 169 the full word is used again).

We agree that every abbreviation should be explained. We changed the text accordingly and added an abbreviation list. We opted to remove HC and wrote ‘honeycombing’ always in full.

There are some minor grammatical/spelling errors throughout the manuscript. For instance:
- line 169: ‘conferred a mean’ instead of ‘conferred an mean’.

- line 199: ‘as’ instead of ‘than’.

- line 245: ‘shown’ instead of ‘showed’.

We thank the reviewer for his comments and changed the text accordingly

Reviewer 2 Report

This is an interesting work about a relevant aspect of fibrotic ILDs.

I have only some minor comments:

Many abbreviations are used in the text without the extensive forms (CTD, fHP, NSIP, UILD, HC, ecc)

Page 3, line 118: INPULSIS is incorrect

Please provide the extensive form of HC the first time in the text and therefore use always the abbreviation

What does it mean GAP system? I suggest to cite also the work of Torrisi et al and the TORVAN model

About the role of GAP-system, i suggest to cite also the reference harari S et al. Clin Respir J 2019

Page 5, line 210: Conclusion: Are CT variables (HC and fibrosis) associated with disease progression also in CPFE?

Page 5, line 245: I suggest to add some informations also about the possible role of pentraxin in IPF and fILDs

I think that some tables and/or figures may add value to this work

Author Response

This is an interesting work about a relevant aspect of fibrotic interstitial lung diseases.

I have some minor comments:

Many abbreviations are used in the text without the use of extensive forms (CTD, fHP, NSIP, UILD, HC, ecc)

We agree that every abbreviation should be explained. We changed the text accordingly and added an abbreviation list.

Pag.3 line118, INPULSIS is incorrect

We thank the reviewer for this comment and removed the typo.

Please provide the extensive form of HC the first time in the text and therefore use always the abbreviation

We opted to remove HC and wrote ‘honeycombing’ always in full.

What does it mean GAP-system? I suggest to cite also the work of Torrisi et al and the TORVAN model.

We changed GAP ‘system’ to ‘GAP score’ which might be more accurate.

We acknowledge that TORVAN – by including comorbidities – is superior in predicting survival in IPF compared to GAP. However, we introduced the GAP system in the manuscript as many studies evaluated the GAP system in nonIPF fIILD, like UILD and CTD-ILD. Unfortunately, to our knowledge, the TORVAN model was not evaluated in nonIPF fILD. We mentioned that GAP has been designed to predict survival and not progression, as parameters (e.g. age) are included which predict the former but not the latter. A fortiori, this is also the case with the TORVAN model, as it includes also comorbidities. Including comorbidities enable better survival prediction, but probably does not improve prediction of disease progression.

About the role of the GAP system, I suggest to cite also the reference: Harari S et al. Clin Respir J 2019

We added the suggested reference to the paper.

Pag. 5, line 210. Conclusion: CT variables (honeycombing and fibrosis extent) are associated with disease progression also in CPFE?

In 2 CPFE cohort studies, which included 50% of nonUIIP cases, presence of honeycombing and fibrosis extent did confer worse survival (Alsumrain Resp Med 2019 and Choi Plos One 2014). We added to the paper:

“The association between honeycombing and fibrosis extent with mortality was confirmed in 2 CPFE cohorts as well [35-36].”

Page 5, line 245. I suggest to add also some information about the possible role of pentraxin in IPF and fibrotic ILDs

We thank the reviewer for his suggestion. The differentiation from monocytes into fibrocytes and profibrotic macrophages was indeed missing. We added to the text:

“Finally, the profibrotic role of macrophages with an M2 phenotype is established.”

and

“Finally, mitigating the differentiation from monocytes into M2 macrophages by recombinant human pentraxin-2 shows a very promising therapeutic option.”

I think that some tables or figures may add value to the work.

We added a panel with key messages to the text:

  • “The PF-ILD has been conceptualized as an important subgroup of nonIPF fILD patients portend dismal outcomes similar to IPF, despite therapeutically addressing the alleged triggering event.
  • Clinical studies have shown antifibrotics to be effective in nonIPF fILD patients with former disease progression. However, former progression has proven to be a rather moderate predictor of future progression; moreover, loss of time and pulmonary function is inherent before the PF-ILD phenotype can be endorsed.
  • Clinical variables (e.g. radiological disease extent and honeycombing presence) might perform better as predictor of progression and can be assessed at time of diagnosis.
  • In the future, predicting progression will ultimately inform the need for (additional) treatment, while assessing the progression-driving disease mechanisms will guide treatment choices.”